# The First Report of a Pulmonary Abscess Due to *Streptococcus intermedius* in Rabbits in Romania

**DOI:** 10.3390/microorganisms13040769

**Published:** 2025-03-28

**Authors:** Vlad Iorgoni, Ionica Iancu, Ionela Popa, Alexandru Gligor, Gabriel Orghici, Bogdan Sicoe, Corina Badea, Cristian Dreghiciu, Iasmina Luca, Paula Nistor, Janos Degi, Luminita Costinar, Corina Pascu, Viorel Herman

**Affiliations:** 1Department of Infectious Diseases and Preventive Medicine, Faculty of Veterinary Medicine, University of Life Sciences “King Mihai I” from Timişoara, 300645 Timişoara, Romania; vlad.iorgoni@usvt.ro (V.I.); alexandru.gligor@usvt.ro (A.G.); corina.badea@usvt.ro (C.B.); paula.nistor@usvt.ro (P.N.); janosdegi@usvt.ro (J.D.); luminita.costinar@usvt.ro (L.C.); corinapascu@usvt.ro (C.P.); 2Department of Semiology, Faculty of Veterinary Medicine, University of Life Sciences “King Mihai I” from Timişoara, 300645 Timişoara, Romania; ionela.popa@usvt.ro; 3Department of Veterinary Emergencies, Faculty of Veterinary Medicine, University of Life Sciences “King Mihai I” from Timişoara, 300645 Timişoara, Romania; gabriel.orghici@usvt.ro; 4Department of Radiology and Imaging, Faculty of Veterinary Medicine, University of Life Sciences “King Mihai I” from Timişoara, 300645 Timişoara, Romania; bogdan.sicoe@usvt.ro; 5Department of Parasitology, University of Life Sciences “King Mihai I” from Timişoara, 300645 Timişoara, Romania; cristian.dreghiciu@usvt.ro; 6Department of Pathological Anatomy and Forensic Medicine, Faculty of Veterinary Medicine, University of Life Sciences “King Mihai I” from Timişoara, 300645 Timişoara, Romania; iasmina.luca@usvt.ro

**Keywords:** *Streptococcus intermedius*, rabbit, lung abscess, pyothorax, antibiotic resistance

## Abstract

*Streptococcus intermedius* is a Gram-positive coccus usually found in the normal digestive or respiratory flora of humans and in some animal species, including rabbits. In conditions of immunosuppression, it can cause serious infections that can be difficult to treat or can even lead to death if not treated properly. *S. intermedius*-induced infections must be taken seriously, and proper treatment needs to be provided as soon as the patient is diagnosed, because otherwise, these infections can evolve in such a dramatic way as to result in the death of the patient. This study reports the case of a young 5-month-old rabbit that was kept in good living conditions by its owner; however, the rabbit developed a respiratory infection that was not treated properly and was at first ignored by the owner. Due to the poor management, the infection became serious and ultimately ended with the death of the animal. The infection was caused by *Streptococcus intermedius* and affected the lungs, in which a large lung abscess developed, eventually leading to the animal’s death. Even though the etiological agent is a commensal of the digestive and respiratory flora, it can induce lethal infections in some cases.

## 1. Introduction

*Streptococcus intermedius* is a Gram-positive, beta-hemolytic coccus belonging to the *Streptococcus anginosus* group, also known as the *Streptococcus milleri* group, which includes *Streptococcus anginosus, Streptococcus constellatus*, and *Streptococcus intermedius* [1,2]. These bacteria are part of the normal flora in the oral cavity, respiratory tract, gastrointestinal tract, and genitourinary tract in humans [3,4,5,6]. While typically commensal, *S. intermedius* can act as an opportunistic pathogen, leading to serious infections, including abscess formation.

Infections caused by *S. intermedius* are notably severe compared to those caused by other members of the group, such as *S. anginosus* and *S. constellatus* [7]. Although *S. intermedius* infections are more commonly reported in humans, they can also occur in animals, including rabbits. In the case of a young male German Pied Giant rabbit, severe respiratory symptoms and a progressive decline in health led to its death. A necropsy revealed a significant pulmonary abscess in the caudal lobe of the left lung, measuring approximately 3.7 cm in diameter, along with extensive pleural lesions. The histopathological examination showed intramural lymphohistiocytic infiltrates, bronchial epithelium thickening, and multifocal liquefactive necrosis.

Although *Pasteurella multocida*, *Staphylococcus aureus*, and *Bordetella bronchiseptica* are recognized as the most frequently involved etiological agents in respiratory infections in rabbits, *Streptococcus intermedius* should also be considered, as it has the potential to cause severe respiratory infections [8,9,10,11,12,13,14].

The identification of *S. intermedius* as the causative agent was confirmed using MALDI-TOF mass spectrometry, and susceptibility testing showed resistance to penicillin, erythromycin, clindamycin, and tetracycline, while sensitivity to cefotaxime, ceftriaxone, levofloxacin, moxifloxacin, and chloramphenicol was noted.

*Streptococcus intermedius* is recognized for its ability to form deep abscesses and pose serious health risks, potentially leading to severe outcomes if not properly managed [2,3,7,15,16,17]. The pathogen’s capability to cause abscesses in both humans and animals underscores the need for vigilant diagnostic and therapeutic strategies. Past cases have documented similar severe infections in humans, with *S. intermedius* found in pulmonary and other abscesses [1,17]. The resistance patterns observed in this case, with contrasts to other strains, highlight the variability in antimicrobial susceptibility and the importance of tailored treatment approaches [16,18,19,20,21].

## 2. Case Study

In this study, a case involving a young male rabbit of the German Pied Giant breed, aged 5 months and weighing 3.65 kg, was examined. The rabbit was owned by a hobby breeder from the western region of Romania, who practiced extensive rabbit breeding. The rabbit was initially purchased at the age of 2 months from another local breeder engaged in similar extensive practices. Initially, the rabbit exhibited normal adaptation, demonstrating healthy growth and weight gain. It was active, consumed food normally, and displayed all the characteristics of a healthy rabbit, as reported by the owner. The rabbit was housed indoors in a room specifically designed for raising rabbits, featuring proper ventilation and controlled humidity levels. The environmental conditions did not appear conducive to the development of a respiratory infection, particularly given the season during which the rabbit fell ill.

However, approximately one month prior to its death, the rabbit began to exhibit reduced daily activity, increased apathy, and signs of respiratory distress. Additionally, a noticeable decline in daily food intake was observed. Despite these symptoms, the owner did not initially perceive a significant health threat, as these changes coincided with the warm season, a period typically associated with decreased activity and weight gain in rabbits.

Two weeks after the onset of clinical signs, the rabbit’s overall condition progressively worsened, with symptoms becoming increasingly severe. At this point, the owner’s veterinarian administered antibiotics along with a vitamin complex in the rabbit’s water supply. Despite these interventions, the rabbit’s condition did not improve, and it continued to deteriorate until it was eventually found deceased by the owner. In the final days, the rabbit exhibited frequent sneezing, severe respiratory difficulties, and a whitish nasal discharge.

In the final days of the rabbit’s life, its symptoms escalated, resulting in a significant reduction in appetite, with the rabbit nearly ceasing to eat entirely. Each attempt to consume food was accompanied by noticeable respiratory distress, and the rabbit rapidly lost weight.

In the same room, 48 other rabbits of varying ages, ranging from one month to three years, were kept, none of which displayed any signs of illness.

Following the rabbit’s death, the owner brought the animal to the Faculty of Veterinary Medicine in Timișoara, where a necropsy was performed at the Department of Infectious Diseases to determine the cause of death. The general examination revealed only minor whitish secretions in the nostrils and significant weight loss, from 5.1 kg to 3.65 kg, indicative of cachexia. The rabbit exhibited severe weight loss (approximately 28.43%) within a month, indicating a serious pathological process likely associated with respiratory infection and pulmonary abscess. The progressive decline correlated with worsening symptoms, with minimal food intake in the final days. This marked weight reduction, combined with pulmonary lesions, suggests a severe bacterial infection. Although no other rabbits exhibited symptoms, the substantial weight loss raises concerns about a potentially transmissible disease, warranting preventive measures. Upon further examination during the necropsy, a small amount of foam was observed in the trachea.

The chest cavity exhibited the most obvious lesions, and a considerable amount of serosanguinolent liquid with fibrin deposits was discovered. Both lungs had adhesions to the walls of the chest cavity and were covered with fibrin. The lungs also had adhesions to the pericardium. There were extensive lesions in the chest cavity and a massive abscess; the pulmonary abscess was localized in the caudal lobe of the left lung, exhibiting a pasty consistency and measuring approximately 3.7 cm in diameter (Figure 1).

No abnormal findings were present in the abdominal cavity, and only the liver was slightly enlarged.

A histopathological examination of the bronchi revealed an interstitial lymphohistiocytic infiltrate characterized by a dense accumulation of lymphocytes and histiocytes within the bronchial walls. There was marked thickening and partial desquamation of the bronchial epithelium (Figure 2). Multifocal liquefactive necrosis, intermixed with clusters of lymphocytes, was observed (Figure 3). Additionally, purulent bronchial exudate, containing numerous neutrophils and macrophages, was present, contributing to the thickening and partial desquamation of the bronchial wall (Figure 4). An area of liquefactive necrosis was demarcated by a lymphohistiocytic infiltrate, predominantly composed of lymphocytes (Figure 5).

The lung parenchyma exhibited extensive fibrinous adhesions to the chest cavity walls and to the pericardium. Histopathological analysis of the lung tissue showed widespread liquefactive necrosis and thickening with lymphocytic and macrophage infiltration of the pleura, as well as alveolar capillary ectasia (Figure 6).

Based on the clinical presentation and necropsy findings of the rabbit, several respiratory pathogens commonly affecting rabbits were considered, including *P. multocida, B. bronchiseptica*, and *S. aureus*. The clinical signs observed, including decreased activity, apathetic behavior, respiratory distress, nasal discharge, and rapid weight loss, combined with the necropsy findings of severe pulmonary lesions such as abscess formation and fibrinous adhesions, suggested a range of potential etiological agents. Among these, *P. multocida* is a well-recognized cause of severe pneumonia, rhinitis, and abscesses in rabbits. *B. bronchiseptica* is also known for causing respiratory infections, characterized by bronchopneumonia, although its specific symptoms were not prominently observed in this case. *S. aureus* can cause significant respiratory issues and systemic abscesses in rabbits, adding to the differential diagnosis [10,11,12,13].

To accurately identify the etiological agent responsible for the animal’s death, samples were meticulously collected from the pulmonary region, specifically from the site of the abscess, as well as from the bone marrow of the femur. The lung specimens, obtained directly from the abscess, were analyzed to determine the presence of any pathogenic microorganisms contributing to the observed pulmonary lesions. Additionally, bone marrow samples from the femur were included to investigate any potential systemic involvement or dissemination of the infectious agent. This comprehensive sampling approach was essential for the precise determination of the causative pathogen and to elucidate the underlying etiology of the fatal condition observed in the rabbit.

Cultures from both the pulmonary tissue and bone marrow yielded pure growth, with *S. intermedius* being the sole bacterium isolated. The specimen was cultured using an agar-based medium enriched with 5% sterile sheep blood, which provided the necessary nutrients and suitable conditions for its growth. The cultures were incubated at 37 °C in a 5% CO_2_; atmosphere for 24 h to ensure optimal growth. The isolation of *S. intermedius* from these sources confirms its role as the causative agent in this case. Quality control was conducted using *Streptococcus intermedius* ATCC 27335. The susceptibility result from the quality control strain fell within the specified quality control ranges [22,23].

Before conducting susceptibility testing, the presumed *Streptococcus* spp. strains were identified using matrix-assisted laser desorption–ionization time-of-flight mass spectrometry (MALDI-TOF MS, Bruker Daltonik, Bremen, Germany). The identification process involved preparing the bacterial samples with an ethanol/formic acid protocol. Specifically, 1 μL of bacterial protein suspension was applied to a MALDI target plate, followed by the addition of 1 μL of matrix solution (10 mg α-cyano-4-hydroxycinnamic acid per mL in 50% acetonitrile and 2.5% trifluoroacetic acid). Bacterial mass spectra were acquired using the Microflex™ mass spectrometer (Bruker Daltonik) and analyzed using the MALDI BioTyper™ 3.0 software package. Identification was performed based on a comparison with the manufacturer’s database, applying Bruker’s standard criteria: scores ≥ 2.0 indicated species-level identification, while scores between 1.7 and 2.0 suggested genus-level identification [8]. The MALDI-TOF analysis confirmed that *Streptococcus intermedius* was the etiological agent of the infection leading to the rabbit’s death. Subsequent testing with the VITEK 2 COMPACT system revealed that the isolated *Streptococcus* strain was susceptible to cefotaxime, ceftriaxone, levofloxacin, moxifloxacin, and chloramphenicol, but resistant to penicillin, erythromycin, clindamycin, and tetracycline [8].

## 3. Discussion

The possible route of infection in this case remains uncertain, but several hypotheses can be considered. One potential source is the introduction of new animals into the breeding group, which could have facilitated the transmission of *Streptococcus intermedius*. Another plausible scenario involves the participation of some rabbits from the facility in profile exhibitions, where they may have been exposed to infected carriers. Additionally, an indirect route of transmission cannot be ruled out, as a family member was recently hospitalized, raising the possibility of bacterial introduction through fomites. Furthermore, given the opportunistic nature of *S. intermedius*, it is likely that other rabbits in the facility may be asymptomatic carriers, acting as potential reservoirs for the pathogen without exhibiting clinical signs. These factors highlight the importance of regular microbiological surveillance and strict biosecurity measures to prevent similar cases in breeding environments.

To prevent future outbreaks, regular surveillance through microbiological screening of both symptomatic and asymptomatic animals, alongside environmental monitoring, is essential. Biosecurity measures, such as controlled animal introductions and quarantine, should also be enforced.

A multi-germ autogenous vaccine, tailored to the bacterial strains present in the facility, could be an effective preventive strategy. Further studies are needed to assess its efficacy in similar settings.

*Streptococcus intermedius* is recognized as a significant pathogen responsible for abscess formation in humans, and evidence suggests its ability to induce severe abscesses in rabbits as well. This bacterium is a common constituent of both human gastrointestinal and oropharyngeal microflora and constitutes part of the normal microflora in rabbits [15,16]. Members of the *Streptococcus anginosus* group, including *S. intermedius*, are known for their propensity to cause deep-seated abscesses, presenting considerable health risks that can lead to mortality if inadequately managed [2,3,17].

In the context of pet rabbits, *S. intermedius*, along with other streptococci of the same group, such as *Streptococcus anginosus*, has been implicated in dental abscesses. Notably, these infections occur in isolation without the concurrent isolation of additional pathogens [16]. *S. intermedius* is frequently associated with cerebral symptoms, hepatic abscesses, and thoracic empyema, with affected individuals often experiencing extended hospitalizations and increased mortality rates compared to infections caused by other members of the *Streptococcus anginosus* group. Predisposing factors for *S. intermedius* infections include dental procedures and sinusitis [1,3,7,16,18,19,20].

A 2017 case study described a 29-year-old male presenting with chest pain and respiratory symptoms. Despite normal blood tests, imaging revealed a 3 cm nodule at the base of the right lung, with subsequent CT scans identifying multiple pulmonary nodules with cavitary features. The biopsy results confirmed *S. intermedius* as the causative agent, with the strain being sensitive to ceftriaxone [1]. Another case reported in 2023 involved a 77-year-old female with Parkinson’s disease and a significant smoking history. This patient experienced respiratory difficulties, and CT scans revealed two circumscribed masses in the thoracic cavity. A biopsy and a bacterial culture identified *S. intermedius* as the pathogen, which was effectively treated with amoxicillin and clavulanic acid [17].

Antibiogram results indicated that the isolated strain of *S. intermedius* exhibited resistance to penicillin, clindamycin, and tetracycline, whereas other strains from different studies showed sensitivity to these antibiotics [16]. Additionally, strains isolated from dental abscesses in both humans and rabbits were found to be sensitive to ceftriaxone, chloramphenicol, cefazolin, ciprofloxacin, tetracycline, and azithromycin, while showing resistance to trimethoprim–sulfamethoxazole and metronidazole [16,21].

In 2023, a case was reported in Italy of a 40-year-old male presenting with persistent cough, fever, and lumbar pain. Imaging revealed a necrotic brain lesion and a consolidative lung abscess. *S. intermedius* was identified as the causative pathogen. The initial treatment included broad-spectrum antibiotics, which were later adjusted based on culture results. The patient underwent surgical drainage of the brain abscess and received targeted therapy with ceftriaxone and dexamethasone. Following a four-week course of antibiotics, the patient showed significant improvement and was discharged in a stable condition. Subsequent follow-ups confirmed the resolution of symptoms and no relapse of infection [19].

In a 2022 study, *Streptococcus intermedius*, a bacterium commonly found in the oral cavity, was highlighted as a potentially dangerous pathogen, particularly when infections are not treated promptly and effectively. The study noted that *S. intermedius* can cause severe infections like liver and brain abscesses due to its virulence factors, such as intermedilysin, which destroys human cells. If left untreated, infections can lead to septicemia and death, underscoring the critical importance of early and appropriate medical intervention to prevent these life-threatening complications [24].

Recent studies highlight a rise in intracranial infections caused by *Streptococcus anginosus* group (SAG) organisms, particularly *S. intermedius*, compared to other bacterial pathogens. These cases are often associated with more surgical interventions and complications, especially in children, who are typically older when infected with SAG organisms. The increase in the incidence of intracranial infections could be due to improved identification methods and potential virulence factors like hydrolytic enzymes and the intermedilysin (ILY) toxin. However, factors such as demographic changes, increased carriage, and alterations in predisposing infections may also contribute, warranting further investigation to understand their combined impact on *S. intermedius* intracranial infection trends [25].

## 4. Conclusions

This case study of a young German Pied Giant rabbit with a severe pulmonary abscess caused by *Streptococcus intermedius* underscores the significant pathogenic potential of the bacterium in both humans and animals. The rabbit’s clinical presentation, including severe respiratory symptoms and rapid deterioration, led to the discovery of a substantial abscess in the left lung’s caudal lobe. Histopathological findings revealed extensive lung damage and inflammatory responses consistent with severe bacterial infection. The identification of *S. intermedius* was confirmed through MALDI-TOF mass spectrometry, with susceptibility testing showing resistance to several antibiotics commonly used for treating such infections, highlighting the need for targeted therapeutic strategies. This case emphasizes the importance of considering *S. intermedius* in differential diagnoses for respiratory infections in rabbits and reinforces the need for effective management protocols. The variability in antibiotic resistance patterns observed in this study, compared to other reported strains, indicates a need for ongoing surveillance and the adaptation of treatment approaches. Overall, this case contributes valuable insights into the clinical and microbiological aspects of *S. intermedius* infections in rabbits and calls for further research to better understand and address these infections.

## Figures and Tables

**Figure 1 microorganisms-13-00769-f001:**
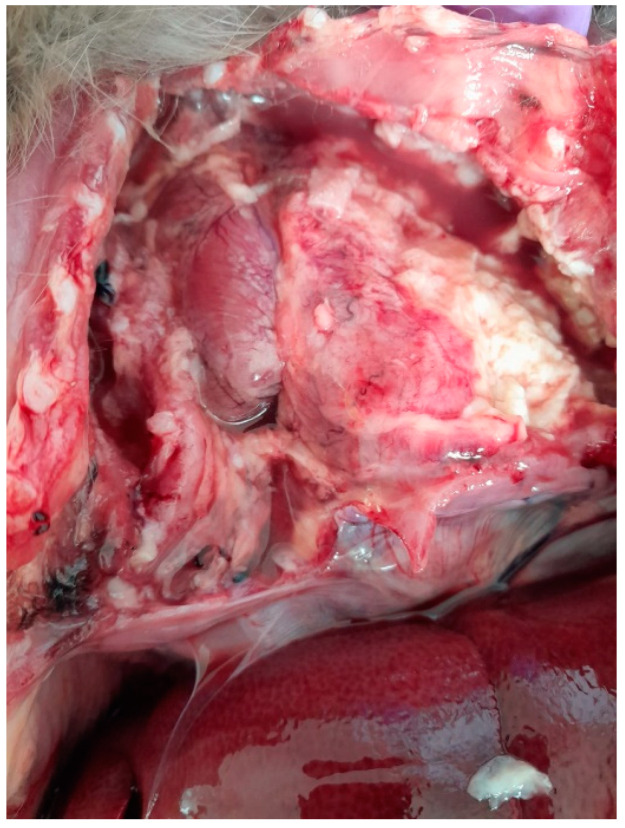
Lung abscess, adhesions to the walls of the chest cavity, with fluid and fibrin in the chest cavity.

**Figure 2 microorganisms-13-00769-f002:**
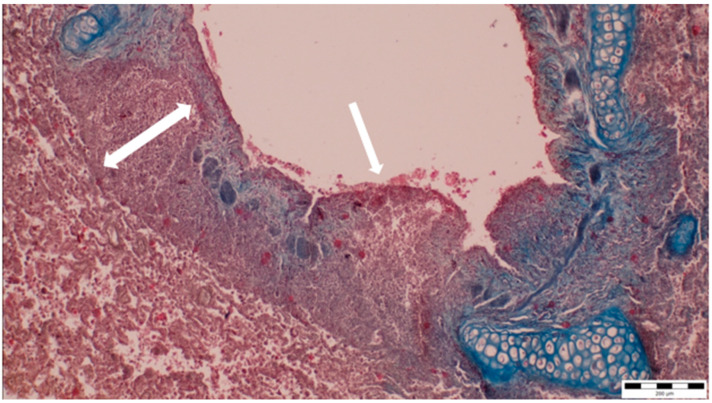
Interstitial lymphohistiocytic infiltrate, along with thickening (two-way arrow) and partial shedding of the bronchial epithelium (white arrow) (obj. ×10).

**Figure 3 microorganisms-13-00769-f003:**
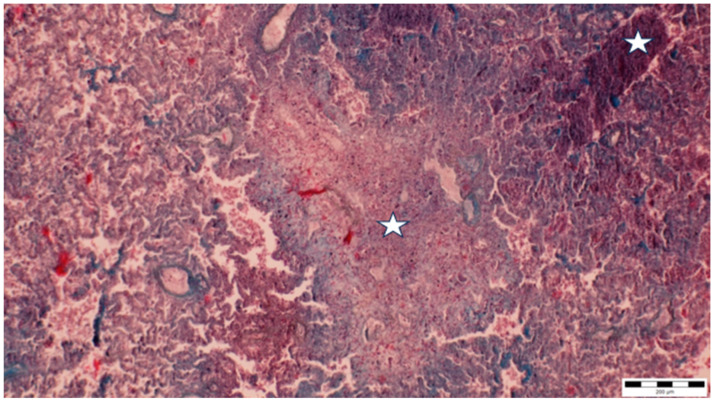
Multifocal liquefactive necrosis intermixed with clusters of lymphocytes (asterisks) (obj. ×10).

**Figure 4 microorganisms-13-00769-f004:**
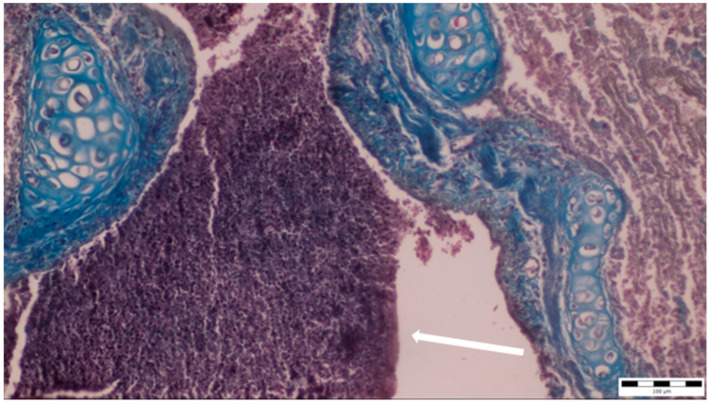
Purulent bronchial exudate containing numerous neutrophils and macrophages was observed (white arrow), alongside thickening and partial desquamation of the bronchial wall (obj. ×20).

**Figure 5 microorganisms-13-00769-f005:**
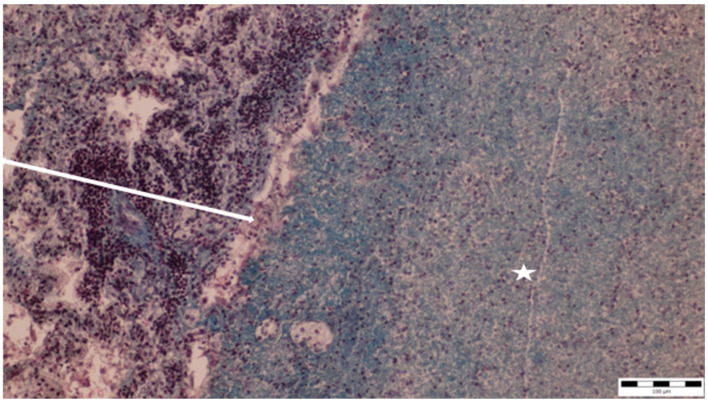
An area of liquefactive necrosis (asterisk) is demarcated at the margins by a lymphohistiocytic infiltrate, primarily composed of lymphocytes (two-way arrow) (obj. ×20).

**Figure 6 microorganisms-13-00769-f006:**
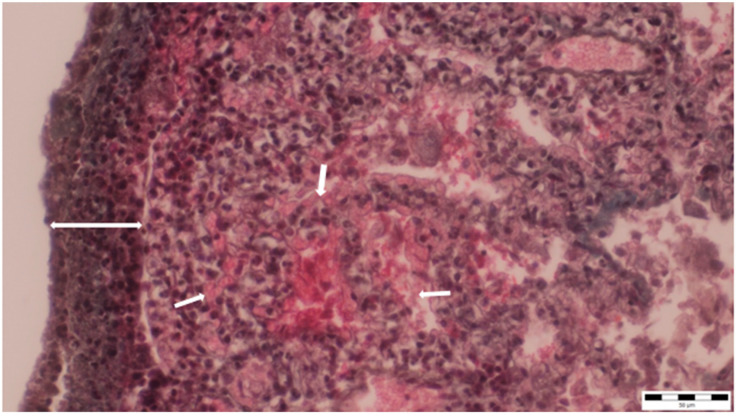
Thickening and infiltration of the pleura with lymphocytes and macrophages (two-way arrow), as well as alveolar capillary ectasia, were observed (small arrows) (obj. ×40).

## Data Availability

The original contributions presented in this study are included in the article. Further inquiries can be directed to the corresponding authors.

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
