# Peer review of "The First Report of a Pulmonary Abscess Due to Streptococcus intermedius in Rabbits in Romania"

_microorganisms, 2025, doi:10.3390/microorganisms13040769_

Round 1
Reviewer 1 Report
Comments and Suggestions for Authors
- Authors did not obtain ethical approval for conducting animal experiments.
- Although it is a case study, it is necessary to explain the experimental methodology.
- The histopathological figures need to have more explanation, by adding arrows to show effected tissues.
- There are some comments on the PDF file.

Author Response
Honorable Reviewer,
We sincerely appreciate your time and valuable feedback on our manuscript. Below, we have addressed each of your comments point by point.
Comment 1: "Authors did not obtain ethical approval for conducting animal experiments."
Response 1: We would like to clarify that no experiments were conducted in this study. The presented results were obtained through investigations performed on a deceased animal. Therefore, ethical approval for conducting live animal experiments was not required. However, to ensure compliance with ethical considerations, we have submitted the approval from the Bioethics Committee to the Editor.
Comment 2: "Although it is a case study, it is necessary to explain the experimental methodology."
Response 2: Thank you for your comment. We would like to clarify that our study did not involve any experiments, as all results were obtained from investigations conducted on a cadaver. No live animal procedures were performed. Regarding the methodology, descriptions of the applied techniques, including post-mortem examination, microbiological analysis, and histopathological evaluation, are provided in the manuscript.
Comment 3: "The histopathological figures need to have more explanation, by adding arrows to show affected tissues."
Response 3: We appreciate this suggestion. We have now updated the histopathological figures by adding arrows to indicate the affected tissues.
Comment 4: "There are some comments on the PDF file."
Response 4: We have carefully reviewed all the comments provided in the PDF and have addressed each of them accordingly in the revised manuscript. Any necessary corrections and clarifications have been made based on your valuable input.
We appreciate your constructive feedback, which has helped us improve the quality of our manuscript. Thank you for your time and effort in reviewing our work.
Reviewer 2 Report
Comments and Suggestions for Authors
The authors described the first letal isolation Streptococcus intermedius at pulmonary tract in a breeded rabbit in Romania.
I recommend to highlight the possible route of infection of this single animal in the breeeding, and future persectives in terms of suveillance and the administration of a multi-germ autogenous vaccine to prevent a possible outbreak in the farm, as reported in other country like Italy.
Comments on the Quality of English LanguageI strongly recommend to improve the quality of language and check the italics
Author Response
Honorable Reviewer,
We sincerely appreciate your time and valuable feedback on our manuscript. Below, we have addressed each of your comments.
Thank you for your valuable feedback and suggestions. In response to your recommendation, we have revised the manuscript to highlight the possible routes of infection in the breeding facility, including the potential introduction of new animals, participation in exhibitions, and the possibility of asymptomatic carriers. These changes can be found in the Discussion section of the manuscript.
Additionally, we have addressed future perspectives regarding surveillance and the use of a multi-germ autogenous vaccine to prevent potential outbreaks. This information has also been incorporated in the Discussion section.
To improve the language quality, we have utilized the English editing services provided by MDPI (editing code: english-edited 91093). This has helped ensure the manuscript meets the highest standards of clarity and accuracy.
We appreciate your constructive comments, which have significantly improved the quality of the manuscript.
Round 2
Reviewer 1 Report
Comments and Suggestions for Authors
I would like to thank authors for their positive response, I have two comments:
- In line 106, authors added the reduction "loose" percentage not the statistical p-value. If they didn't make statistical analyzes they should remove the word significant from the text and keep the percentage. Kindly check the comment in pdf file.
- In line 177, the word environment is not appropriate for the context, kindly clarify the intended of this word.

Author Response
Honorable Reviewer,
We sincerely appreciate your time and valuable feedback on our manuscript. Below, we have addressed each of your comments point by point.
Comment 1: "In line 106, authors added the reduction 'loose' percentage not the statistical p-value. If they didn't make statistical analyzes they should remove the word significant from the text and keep the percentage. Kindly check the comment in the pdf file."
Response 1: Thank you for your observation. We acknowledge that no statistical analysis was performed; therefore, we have removed the term "significant" and retained only the percentage to accurately reflect the observed weight loss. The revised text now clarifies this point.
Comment 2: "In line 177, the word environment is not appropriate for the context, kindly clarify the intended of this word."
Response 2: We appreciate your suggestion and have replaced the term "environment" with "suitable conditions" to better reflect the intended meaning in the context of bacterial growth. This revision ensures greater clarity and precision.
We appreciate your constructive feedback, which has helped us improve the quality of our manuscript. Thank you for your time and effort in reviewing our work.